



# 1 Gap-Filling of Turbulent Heat Fluxes over Rice–Wheat-Rotation
# 2 Croplands Using the Random Forest Model

Jianbin Zhang[1], Zexia Duan[1], Shaohui Zhou[1], Yubin Li[1], Zhiqiu Gao[1]
[1] School of Atmospheric Physics, Nanjing University of Information Science & Technology, Nanjing, 210044, China
*Correspondence to*: Dr. Yubin Li (liyubin@nuist.edu.cn)
**Abstract.** This study investigated the accuracy of the Random Forest (RF) model in gap-filling the sensible (H) and latent heat
(LE) fluxes, by using the observation data collected at a site over rice–wheat-rotation croplands in Shouxian County of eastern
China from 15 July 2015 to 24 April 2019. Firstly, the variable significances of the machine learning (ML) model's five input
variables, including the net radiation (Rn), winds speed (WS), temperature (T), relative humidity (RH), and air pressure (P),
were examined, and it was found that Rn accounted for 78% and 76% of the total variable significance in H and LE calculating,
respectively, showing that it was the most important input variable. Secondly, the RF model's accuracy with the five-variable
(Rn, WS, T, RH, P) input combination was evaluated, and the results showed that the RF model could reliably gap-fill the H
and LE with mean absolute errors (MAEs) of 5.88 Wm$^{-2}$ and 20.97 Wm$^{-2}$, and root mean square errors (RMSEs) of 10.67 Wm$^{-2}$
$^{2}$ and 29.46 Wm$^{-2}$, respectively. Thirdly, 4-variable input combinations were tested, and it was found that the best input
combination was (Rn, WS, T, P) with the MAE of H and LE reduced by 12.65% and 7.12%, respectively, after removing RH
from the input list. At last, through the Taylor diagram, H and LE gap-filling accuracy of the RF model, the support vector
machine (SVM) model, the k-nearest neighbor (KNN) model, and the gradient boosting decision tree (GBDT) model was
inter-compared, and the statistical metrics showed that RF was the most accurate for both H and LE gap-filling, while the LR
and KNN model performed the worst for H and LE gap-filling, respectively.

## 21 1 Introduction

The turbulent fluxes between the atmosphere and the ground play a crucial role in global climate change and atmospheric
circulation, and the inaccuracy of long-term observations of surface turbulent fluxes is a major factor in erroneous weather
predictions and climate projections. Research on the ecological effects of urban green spaces, agricultural ecosystems, and
forests all use surface turbulent fluxes as key indicators. Currently, the eddy covariance (EC) technique can be used to directly
measure the turbulent fluxes (Wilson et al., 2001; Jiang et al., 2021; Wang et al., 2021). However, due to sensor failure and
adverse meteorological factors (such as rainfall and frost), these high-frequency turbulence data are subject to errors (Khan et
al., 2018). As a result, it is difficult to obtain a continuous time series of ground-based turbulent fluxes. Furthermore, quality
assurance methods lead to unavailable sections of flux datasets (Nisa et al., 2021). Based on the above reasons, gap-filling is
in need to retrieve continuous datasets of EC-based fluxes. Researchers have developed approaches based on existing



meteorological information to fill up the gaps in atmospheric databases, such as interpolation, nonlinear regression, mean
diurnal method, and sampling techniques from the marginal distribution (Falge et al., 2001; Hui et al., 2004; Stauch et al.,
2006; Foltnov et al., 2020). Further, the ML technique has also become an effective method to be used in the calculation of
turbulent fluxes (McCandless et al., 2022).

As a result of recent developments in high computing technology, machine-learning-based algorithms have been developed
and successfully used in various areas, such as natural language processing, data mining, biometrics, computer vision, search
engines, clinical applications, video games, robots, etc. To address the missing data issue, machine-learning-based models
have recently been used to fill data gaps in meteorological elements and turbulent fluxes (Bianco et al., 2019; Yu et al., 2020).
As a result of their reliable and repeatable results, these models are now regarded as a standard gap-filling algorithm (Beringer
et al., 2017; Isaac et al., 2017). ML algorithms have several deficiencies even if they perform well in some areas. For instance,
over-fitting is a major concern that can occur when the training window is too short or the training dataset's quality is poor.
That's because the present ML approaches are not sufficiently adaptable to work in extreme situations with large values
(Kunwor et al., 2017; Moffat et al., 2007). Furthermore, even with the best technique, the model uncertainty of gap-filling still
plays a role, particularly when the gaps are relatively large. Numerous novel ML and optimization algorithms have been created
and put to use in numerous scientific domains since the 2000s, and their superiority has been demonstrated, either singly or as
a component of a hybrid or ensemble model (e.g. Gani et al., 2016).

Based on the need for fluxes dataset gap-filling, and the effectivity of the ML technique, this paper aims to, firstly, investigate
the performance of the RF machine learning algorithm trained from a dataset obtained over rice-wheat-rotation croplands in
Shouxian County, eastern China, in gap-filling the sensible and latent heat fluxes; and secondly, to analyze the RF model's
accuracy with various meteorological input combinations during training; and thirdly, to compare the performance of RF model
with other four typical ML models.

**2 Materials and Methods**
**2.1 Study area**
This observation was conducted at a site in Shouxian County in the eastern Chinese province of Anhui (32.42 °N, 116.76 °E)
(Figure 1). The altitude of the site is 27 meters, and the annual mean air temperature and annual cumulative precipitation here
are 16 °C and 1115 mm, respectively. This observation site is rather flat, with farmland accounting for more than 90% of the
area. Winter wheat is grown here from November until late May, while from June to November the field is flooded, plowed,
and harrowed as rice paddies (Duan et al., 2021) (Figure 2). The subtropical northern boundary of the monsoon humid climatic
type describes the area's climate.




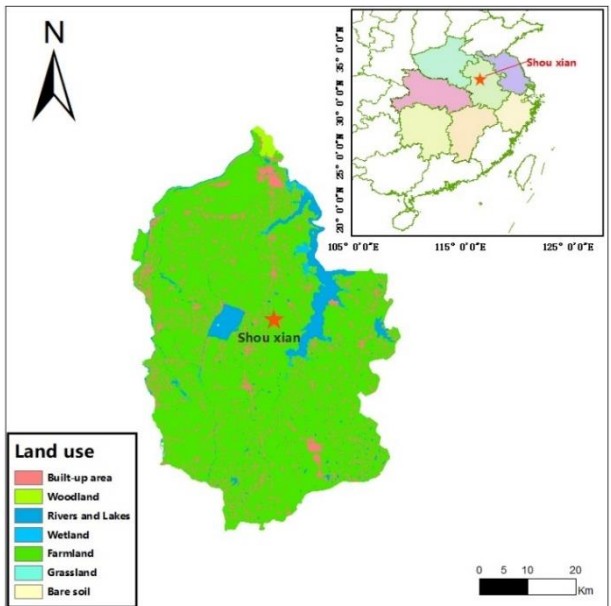


**Figure 1. Geographical location and land-cover map of Shouxian County.**

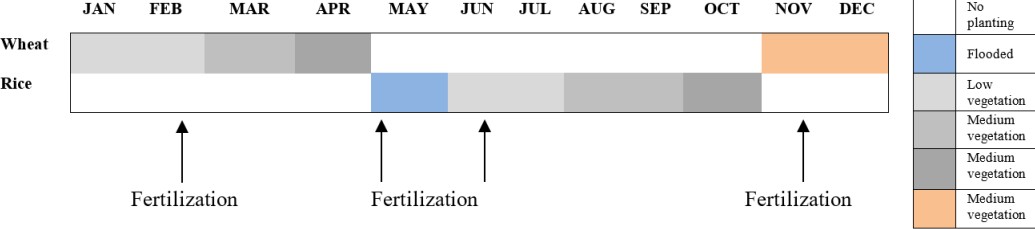


**Figure 2. Crop calendars for the rice and wheat in the North Yangtze River Delta region.**

**2.2 Data**
Over the site described above, EC sensors (EC 150, Campbell Scientific Inc., Logan, UT, USA) were installed at 2.5 meters
above the ground, including a three-dimensional sonic anemometer (CSAT3, Campbell Scientific Inc., Logan, UT, USA) and
a $CO_2/H_2O$ open-path infrared gas analyzer. The sensible and latent heat fluxes were computed half-hourly using EddyPro
software, with time lag compensation, double coordinate rotation, spectrum correction, and Webb-Pearman-Leuning density





correction (Wutzler et al., 2018; Anapalli et al., 2019). Poor-quality fluxes (Eddypro quality check flag value=2) were
discarded. And a quality check based on the relationship between the measured flux and friction velocity was carried out to


**Figure 3. Daily averaged a) Rn: net radiation(Wm$^{-2}$) , b) u\*: friction velocity(m/s), c) T: air temperature(°C), d) RH: relative humidity(%), e) P: air pressure(hPa), and f) WS:wind speed(m s$^{-1}$).**






remove the biased data (Papale et al., 2006). Then, using the marginal distribution sampling technique, the flow data were gap-
filled (Reichstein et al., 2005). The time series of air temperature, relative humidity, wind speed, air pressure, friction velocity,
and net radiation were also subjected to quality control. According to the criteria of $X(h) < (X - 4\sigma)$ or $X(h) > (X + 4\sigma)$, where
$X(h)$ indicates the time series of the component, X is the mean across the averaging interval, and σ is the standard deviation,
noisy data were eliminated (Gao et al., 2003). Data observed from 15 July 2015 to 24 April 2019 are used in this study, and
Figure 3 shows the daily average data of Rn: net radiation($W\ m^{-2}$), u*: friction velocity(m/s), T: air temperature(℃), RH:
relative humidity(%), P: air pressure(hPa), and WS: wind speed($m\ s^{-1}$).

## 2.3 The RF Model

RF is a machine learning method that is quick, adaptable, and frequently used to analyze classification and regression jobs
(Breiman, 2001). This model can successfully evaluate highly dimensional and multicollinear data and is resistant to overfitting
(Belgiu et al., 2016). The RF model provides a feature-selection tool to assist in determining the importance of the predictor.
The contribution of each variable to the model, with important variables having a higher effect on the results of the model
evaluation, is the definition of feature significance (Liu et al., 2021). 90% of the data collected at the Shouxian observation
site throughout the study period were used to train the RF model, while the remaining 10% was used to independently validate
the model (hereafter, validation dataset). To lessen the overfitting in this case, a 10-fold cross-validation (CV) procedure was
used (Cai et al., 2020). All training data used here was randomly divided into ten subsamples of equal size for the 10-fold CV
tests. And nine out of the ten subsamples were used as training data (hereafter, training dataset), while the remaining subsample
was used as testing data (hereafter, testing dataset). All ten of the subsamples were utilized as testing data exactly once for
each of the 10 iterations of the CV procedure. One estimate was created by averaging the 10 findings from the folds. We
modified the four RF model hyperparameters based on Bayesian optimization to get the optimal model (Baareh et al., 2021;
Frazier, P.I., 2018): the maximum number of features considered to split a node (Max features), the maximum number of trees
to build (n estimators), the minimum sample number placed in a node prior to the node being split (min split), and the maximum
number of levels for each decision tree (Max depth). The simulated performance of the 10-fold CV outcomes was evaluated
using four statistical metrics: the correlation coefficient (*r*), mean absolute error (MAE), root mean square error (RMSE), and
standard deviation($\sigma_n$). As a result, the final RF model's parameters were adjusted to n estimators = 246, min split = 2, Max
features = 10, and Max depth = 35, to have the best statistical metrics.
The four statistical metrics are calculated by:

$$r = \frac{\sum_{i=1}^{N}(S_i - \bar{S})(O_i - \bar{O})}{\sqrt{\sum_{i=1}^{N}(S_i - \bar{S})^2}\sqrt{\sum_{i=1}^{N}(O_i - \bar{O})^2}},$$   (1)



$$MAE = \frac{1}{N}\sum_{i=1}^{N}|S_i - O_i|,$$   (2)






$$\text{RMSE} = \sqrt{\frac{\sum_{i=1}^{N}(S_i - O_i)^2}{N}}, \qquad (3)$$


$$\sigma_n = \frac{\sqrt{\sum_{i=1}^{N}(S_i - O_i)^2}}{N}. \qquad (4)$$


where $S$ stands for the modeled value, $O$ is the observation, $\overline{O}$ is the mean observed value, and $\overline{S}$ is the mean modeled
observation, $\sigma_n$ indicates the standard deviation. The subscript $i$ represents the serial number of samples, and $N$ represents the
total number of samples.

**3 Results and discussion**
**3.1 Driving Factors of H and LE on a Seasonal Scale**
The possible driving factors of H and LE were investigated to determine their respective contributions by the RF model as
shown in Figure 4. Rn, which accounted for 78% and 76% of the total variable significance of H and LE, respectively, was the
most crucial variable in regulating the heat fluxes (Figures 4a and 4c). Consistent with the high variable significance values,
H and LE also had the highest $r$ of 0.79 and 0.75 with H and LE, respectively, as shown in Figures 4b and 4d. The other four
factors contributed much smaller than Rn, and WS, T, RH, and P had importance values of 2%, 4%, 7%, and 5% (2.2%, 19%,
2%, and 0.6%) for H (LE), respectively. In general, all of these predictors played a role in the H and LE calculation, and for
H, the sequence of importance was Rn, RH, P, T, and WS; while for LE, it was Rn, T, WS, RH, and P. The most significant
impact on the change of H and LE came from Rn, which was the most important energy source of the surface and modulated
the surface temperature directly. The WS, T, and RH also affected H and LE according to the Monin-Obukhov similarity
theory (Monin and Obukhov, 1954), while P represented the contributions from the background weather systems.







**Figure 4. The feature importance of the variables for a) H and c) LE, and the correlation coefficient between each of the input variables for b) H and d) LE.**

## 3.2 RF Model Evaluation

Figures 5-6 show the comparison between the observed and the RF-estimated H and LE, respectively. In the period of rice, the RF model showed good performance for both the training dataset (MAE =8.51 and 17.89 Wm$^{-2}$; RMSE =14.11 and 29.82 Wm$^{-2}$, for H and LE, respectively) and the testing dataset (MAE =9.61 and 10.34 Wm$^{-2}$, RMSE = 15.63 and 17.21Wm$^{-2}$, for H and LE, respectively) (Figures 5a, 5b, 6a, and 6b). RF model also showed high consistency with direct measurements for the validation dataset (MAE=5.88 and 20.97 Wm$^{-2}$, RMSE = 10.67 and 29.46 Wm$^{-2}$, for H and LE, respectively), (Figures 5c and 6c). In the period of wheat, the performance of the RF model for the training, testing, and validation datasets of H and LE was similar to that in the period of rice. For the training, testing, and validation datasets, respectively, the MAEs are 7.18, 8.01, and 6.01 Wm$^{-2}$ for H, and 13.58, 8.82, and 19.93 Wm$^{-2}$ for LE; and the RMSEs are12.27, 13.61, and 9.86 Wm$^{-2}$ for H, and



24.92, 15.17, and 28.74 Wm$^{-2}$ for LE (Figure 5d,e,f, Figure 6 d,e,f). These results demonstrate that the RF model is capable of
effectively calculating the H and LE with input variables of Rn, WS, T, RH, and P.

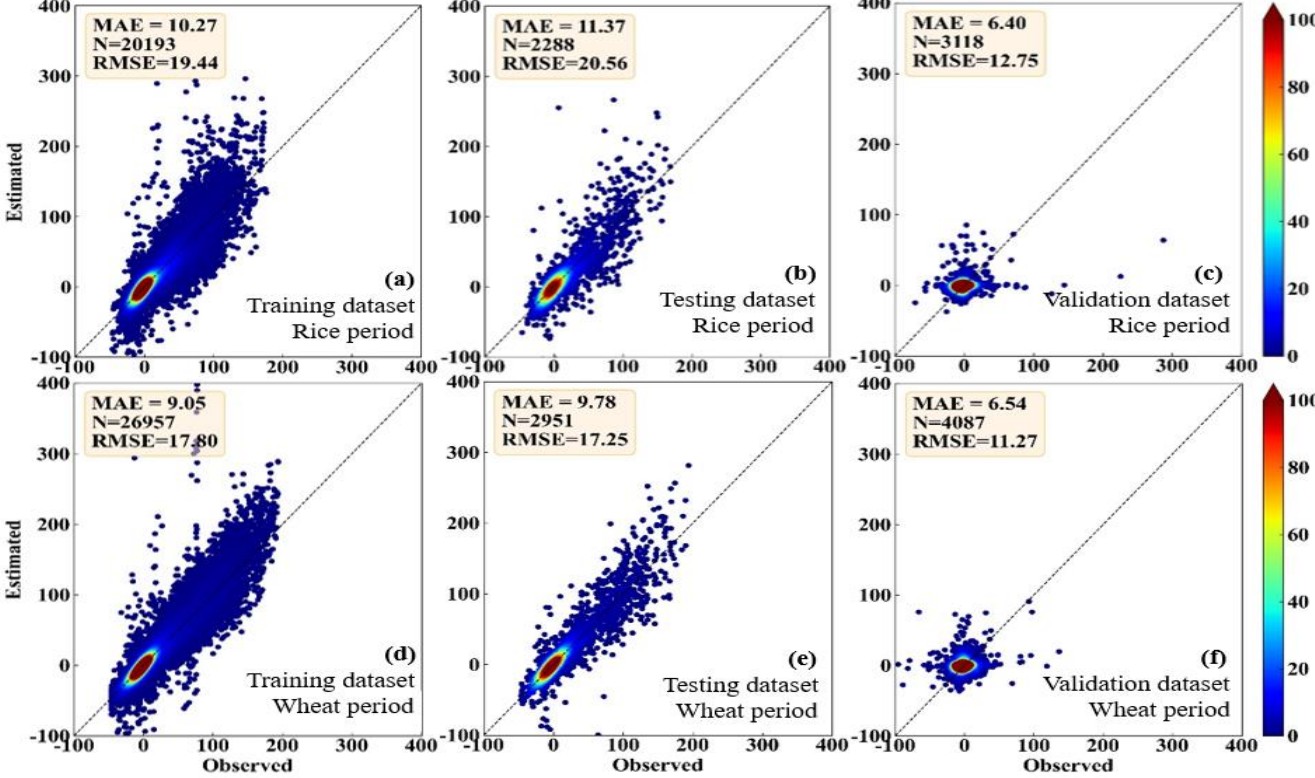



**Figure 5. Scatter density plots of the observed and the RF-estimated H values, a) and d) for the training dataset, b) and e) for the**
**testing dataset, and c) and f) for the validation dataset.  And a), b) and c) are in the period of rice, while d), e) and f) are in the period**
**of wheat.**





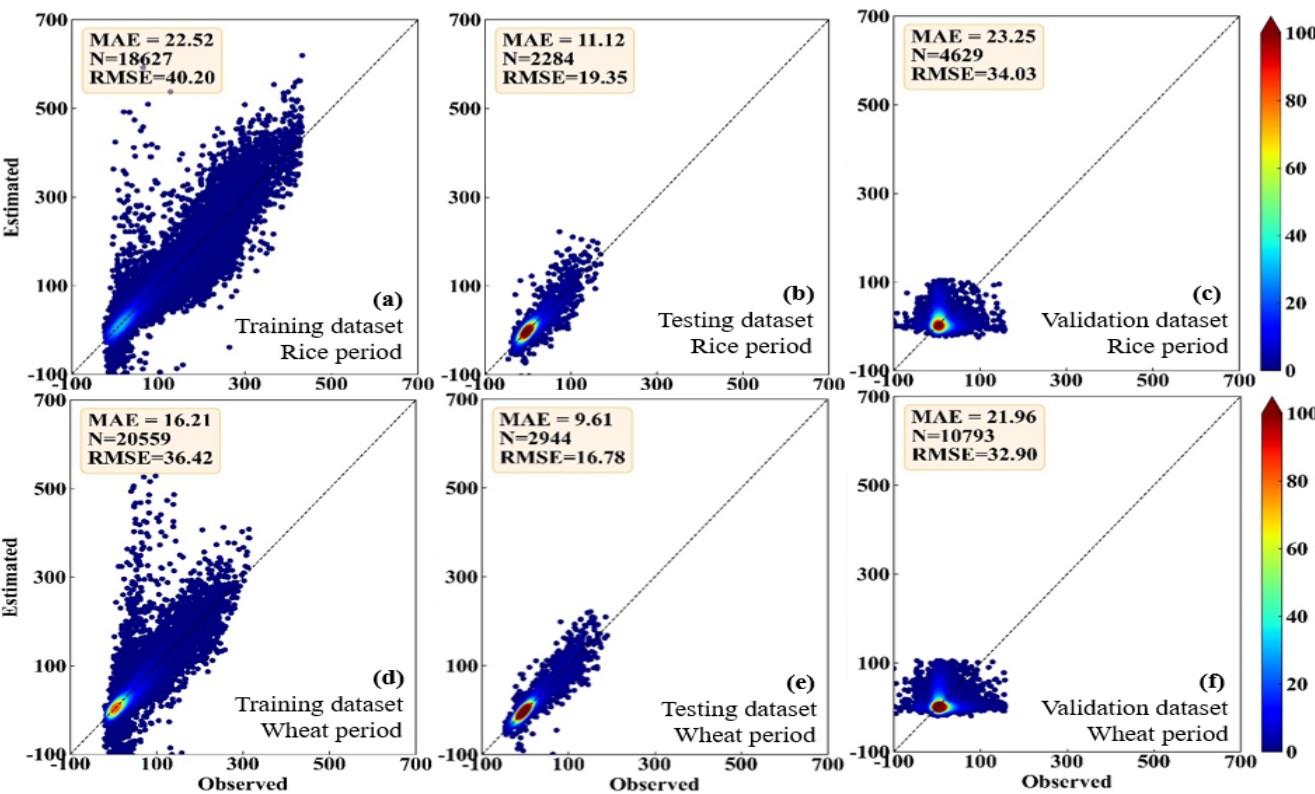

**Figure 6. Same as Figure 5, but for LE.**

**3.3 Examination of Input Combinations**

Meteorological elements may occasionally be unavailable due to the failure of sensors so the 5-variable input combination derived in Section 3.2 is not always applicable. Therefore, examination of other alternative input combinations is important to have substitute choices for data gap-filling when the 5-variable input combination is unavailable. In this subsection, we investigated the RF model's performance under the situation of lacking one element in the 5-variable input combination, i.e., we tested the 4-variable input combinations of (WS, T, RH, P), (Rn, T, RH, P), (Rn, WS, RH, P), (Rn, WS, T, P), and (Rn, WS, T, RH), by removing Rn, WS, T, RH, and P from the 5-variable input combination, respectively. The MAEs and RMSEs for these combinations are shown in Table 1, and it demonstrates that the RF model's accuracy may either increase or decrease as a result of the removal of a meteorological element during the training phase. For instance, it was found that the model's performance greatly improved once RH was eliminated from the input combination, with the MAE and RMSE of H decreasing from 6.48 and 11.94 $Wm^{-2}$ to 5.66 and 11.06 $Wm^{-2}$, respectively, and LE from 19.1 and 39.39 $Wm^{-2}$ to 17.74 and 35.27 $Wm^{-2}$. The results suggested that RH at a single level was not well correlated to the fluxes as shown in Section 3.1, because the





one-level RH was strongly affected by the irrigation activity which was an external factor of the weather system. As a result,
RF model performance was enhanced when the irrelevant variable (i.e., RH) was removed from the input list. The same
condition also happened to the removal of WS, as could be seen from Section 3.1, WS showed small correlations with the
fluxes. WS over this site was rather small, and frequently below 2 m s$^{-1}$, and under this light wind condition, the fluxes were
mostly driven by the buoyancy rather than the wind shear. Figure 7 presents the MAE variation percentage of the 4-variable
input combinations from the 5-variable input combination. After RH was removed from the input list, the RF model showed
favorable performance for both H and LE, as shown in Figure 7, with MAE improvements of 12.65 and 7.12%, respectively.
Notably, the removal of Rn from the input combination resulted in a considerable decline in the RF model's performances,
with MAE degradation percentage values reaching 16.20% and 10.73%, respectively. This outcome makes sense since Rn is
highly associated with H and LE; hence, performance will be declined if Rn is left out of the input training dataset. As a
consequence, our findings demonstrated that choosing strongly associated components could greatly increase the gap-filling
accuracy. According to our findings, the best input combination is (Rn, WS, T, P).

**Table 1.The MAEs and RMSEs of the RF-estimated heat fluxes for the 4-variable input combinations, and the corresponding**
**changes from the 5-variable input combination.**

| Factors Included | Factors Eliminated | | MAE (change) | RMSE (change) |
|---|---|---|---|---|
| **WS, T, RH, P** | Rn | H | 7.63 (+1.15) | 10.72 (−1.22) |
| | | LE | 21.15 (+2.05) | 39.38 (−4.62) |
| **Rn, T, RH, P** | WS | H | 6.15 (−0.33) | 11.42 (−0.52) |
| | | LE | 18.36 (−0.74) | 36.13 (−2.34) |
| **Rn, WS, RH, P** | T | H | 6.68 (+0.20) | 11.48 (−0.46) |
| | | LE | 19.54 (+0.44) | 38.54 (−1.46) |
| **Rn, WS, T, P** | RH | H | 5.66 (−0.82) | 11.06 (−0.88) |
| | | LE | 17.74 (−1.36) | 35.27 (−4.12) |
| **Rn, WS, T, RH** | P | H | 6.49 (+0.03) | 11.77 (−0.17) |
| | | LE | 19.12 (+0.02) | 38.13 (−1.07) |






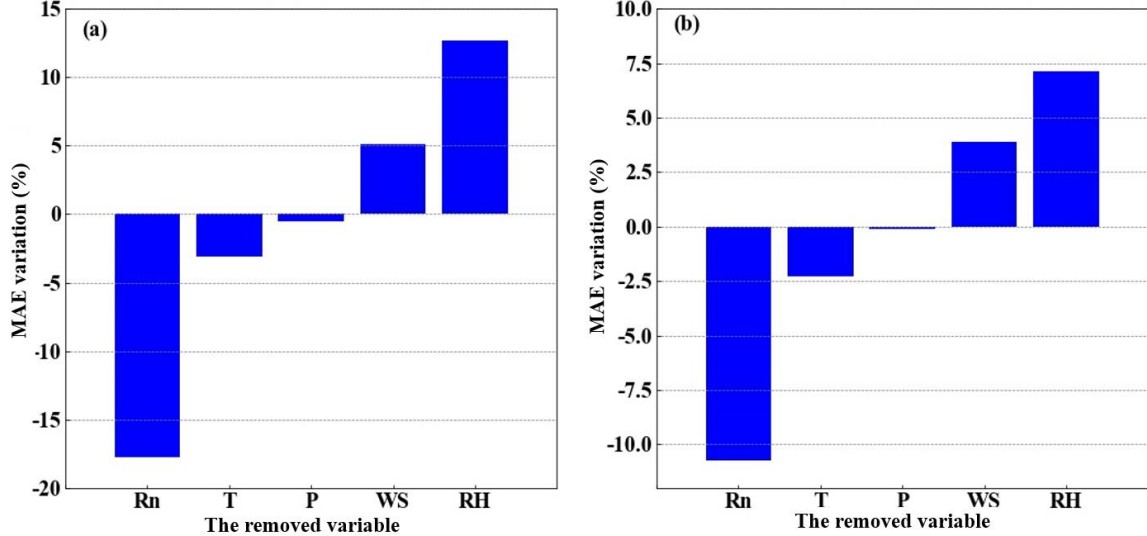

**Figure 7. The MAE percentage variation after changing the 5-variable input combinations to the 4-variable input combinations, a) for H, and b) for LE, respectively. The x-axis labels indicate the removed variables.**

It should be noted that other variables that might have an impact on the H and LE were not investigated here. For example, given that our research site was over farmland and plants were growing, knowledge of the variations of the leaf area index (LAI) and inclusion of it to the training dataset should also be useful to increase the accuracy of the RF model in H and LE gap-filling. The monsoonal climate here also incurred considerable precipitation variations, which might as well potentially contribute to the RF model accuracy improvement. However, due to the lack of LAI and precipitation observations, the inclusion of the two variables into the RF model training dataset was not applicable in this study. Additionally, as shown above, more variables would bring a higher observation demand, and lead to more complexity and potentially decreased results, such as the adding variable of RH.

**3.4 Comparison with other four ML methods**

3.4.1 Comparison in H estimation

To further investigate the reliability of the RF model, we used a Taylor diagram to compare its performance in H estimation with other four ML models: linear regression (LR), k-nearest neighbor (KNN), support vector machine (SVM), and gradient boosting decision tree (GBDT). All the models were optimized with the same technique described above for the RF model. The results are shown in Figure 8. The EC measurements were used as the benchmark. It can be seen that the RF model generally outperforms the other four models, with the standard deviation ($\sigma_n$) and correlation values of 1.05 and 0.98 during





the period of rice planting, and 0.96 and 0.95 during the period of wheat planting, respectively. The SVM model is the second
most accurate model, with the $\sigma_n$ and correlation of 0.92 and 0.98 during the period of rice planting, and 0.91and 0.93 during
the period of wheat planting, respectively. The LR model performs the worst, with the $\sigma_n$ and correlation of 0.60 and 0.76
during the period of rice planting, and 0.80 and 0.72 during the period of wheat planting, respectively. The accuracy of KNN
and the GBDT models is in between the above-discussed models, and the $\sigma_n$ and correlation during the rice and wheat period
for KNN are 0.68 and 0.73, and 0.77 and 0.82; and for GBDT are  0.79 and 0.80, and 0.81 and 0.9, respectively.

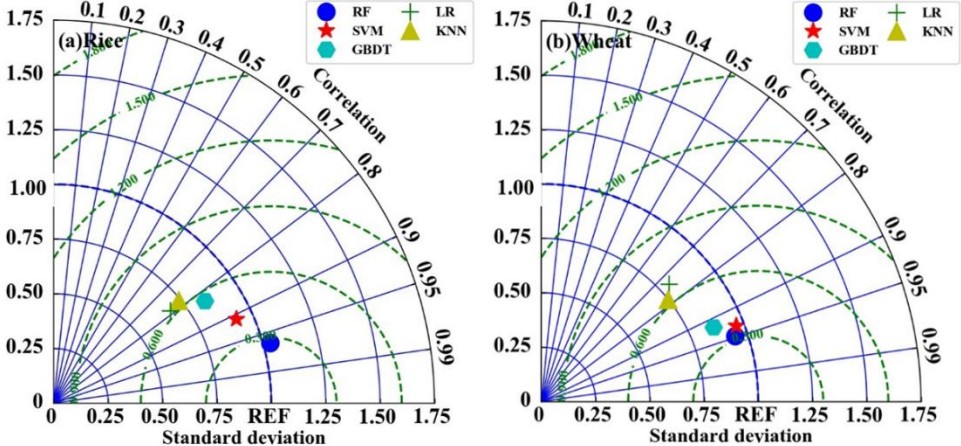


**Figure 8. Taylor diagram visualizing the performances of the five models for estimating H in the period of a) rice and b) wheat.**

3.4.2 Comparison in LE estimation

Figure 9 illustrates a comparison of the estimated LE by all the five models during the period of rice and wheat planting. The
results are similar to those in the H estimation, and the RF model is found to perform better than the other four models, with
the $\sigma_n$ and correlation values of 0.95 and 0.97 during the period of rice planting, and 0.97 and 0.96 during the period of wheat
planting, respectively. Nonetheless, the KNN model performs the worse for LE estimating and has the $\sigma_n$ and correlation
values of 0.68 and 0.82 during the period of rice planting, and 0.62 and 0.79 during the period of wheat planting, respectively.
Overall, as shown by the Taylor diagram of Figures 8 and 9, in this study the RF model has the best accuracy in either H or
LE estimation for data gap-filling.




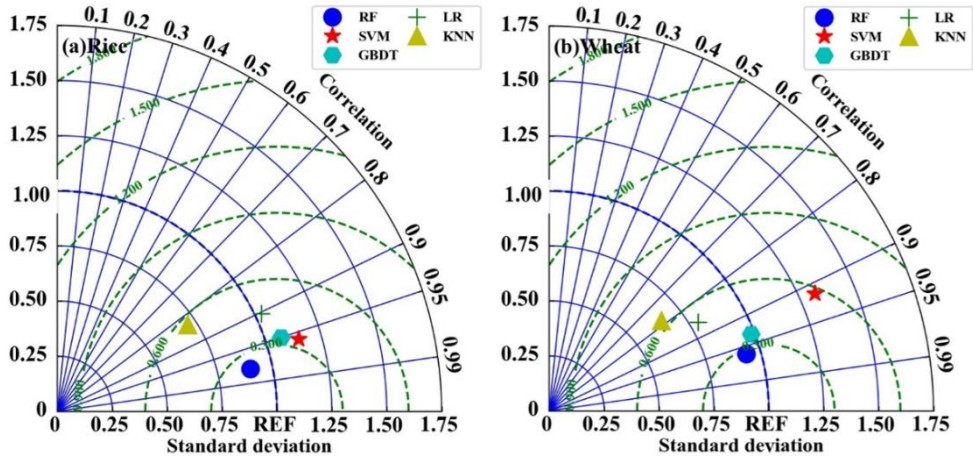

**Figure 9. Same as Figure 8, but for LE**

## 4 Summary and Conclusions

To assess the RF model's capacity for gap-filling the sensible and latent heat flux measurements over rice-wheat rotation croplands, 90% of the total observation data gathered at Shouxian were utilized for training and testing, and the remaining 10% for independent validation. Our findings demonstrate that Rn is the most important variable in regulating H and LE, and it accounts for 78% and 76% of the total variable significance in the RF model construction for H and LE calculation, respectively. The least important variables are WS and P, and their total variable significances are 2% and 0.6%, respectively. During the periods of rice and wheat planting, the RF model with a 5-variable input combination shows reliable performance, with MAE values of 5.88 Wm$^{-2}$ and 20.97 Wm$^{-2}$, and RMSE values of 10.67 Wm$^{-2}$ and 29.46 Wm$^{-2}$, respectively. However, further analysis of the RF model with 4-variable input combinations indicates that the performance of the model is improved when RH is removed from the input list, and the MAE decreases by 12.65% and 7.12% for H and LE, respectively. Nonetheless, the 4- variable input combination without Rn causes an increase in the MAE of the model, by 16.20% and 10.73% for H and LE, respectively. Therefore, the best input combination found in this study for heat fluxes gap-filling is (Rn, WS, T, P). Statistical comparison of RF and other four typical ML models (LR, KNN, SVN, and GBDT) by Tylar diagram further shows that RF is the most accurate, with the $\sigma_n$ and correlation values of 0.95 and 0.97 during the period of rice planting, and 0.97 and 0.96 during the period of wheat planting, respectively. While the LR and KNN models perform the worst for H and LE gap-filling, respectively, according to the statistical metrics of the Tylor diagram.

This study is based on only the data collected over rice–wheat-rotation croplands, but the method presented above to find a reliable heat fluxes gap-filling ML model can also be used over the underlying surface of other types and in other climate



zones. It should be noted that over different types of the underlying surface and climates, the variable significances can vary
and a careful check of the input combinations is needed. For example, over polar oceans with strong winds, Rn probably is not
the most important driving factor, while the winds which cause mostly the turbulence may take the first place. On the other
hand, over areas without human irrigation activity, RH will possibly be strongly related to the latent heat flux, and hence the
inclusion of it into the input list may increase the ML model performance. Besides the examination of the input combinations,
the choice of an ML model and the method to optimize its parameters are also important.

Overall, this study shows the potential to use the RF model to produce trustworthy gap-filling data of H and LE over rice–
wheat-rotation croplands, and the ML methods are suggested to be used to derive the fluxes' estimations when direct EC
observations are not available.

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
