# Peer review of "Gap-Filling of Turbulent Heat Fluxes over Rice—Wheat-Rotation Croplands Using the Random Forest Model"

_Atmospheric Measurement Techniques, 2022_

## Author Response (AR1)

Response to Reviewer 1 and Reviewer 2:

The reviewers give no suggestion in this round, and we are grateful to the reviewers for their positive recognition.

---

## Author Response (AR2)

This paper investigates the performance of machine learning algorithms in gap-filling the sensible and latent heat fluxes over rice-wheat-rotation croplands. The study shows the potential to use the RF model to produce trustworthy gap-filling data of sensible and latent heat fluxes. My detailed comments are as follows:

**Response:** Thank you very much for your positive recognition and helpful suggestions.

**Comment 1:** Please explain the reason of choosing Shouxian County as the study area.

Response: The local climate is characterized as a northern subtropical semi-humid monsoon climate. The annual mean temperature is about 15 ℃ and annual precipitation is about 1115mm. Summer (from June to September) precipitation accounts for nearly 60% of the annual precipitation amount, which meets the high water demand for rice. Drought sometimes occurs due to a lack of precipitation in the growing season of wheat. The terrain of the site is flat and covered with yellow cinnamon soil according to the classification system of the Food and Agriculture Organization. Therefore, this area is a relatively representative research area.

**Comment 2:** I do not see any data gap in Figure 3. Please give a more detailed description of the data used in this study.

Response: Figure 3 shows the inputs for the RF models. including: Rn: net radiation$(Wm^{-2})$ ,u*: friction velocity(m/s), T: air temperature(℃), RH: relative humidity(%),P: air pressure(hPa), WS: wind speed(m/s).The time series of air temperature, relative humidity, wind speed, air pressure, friction velocity, and net radiation were subjected to quality control. The missing data which need gap-filling are H and LE, with 7205 and 16013 missing, accounting for 12.09% and 26.87% respectively.

**Comment 3:** How to prevent overfitting during gap-filling ?

Response: To lessen the overfitting in this case, a 10-fold cross-validation (CV) procedure was used. All training data collected at the Shouxian location for the whole observation period was randomly divided into ten subsamples of equal size for the 10-fold CV tests. The remaining subsample was used as testing data, and nine of the ten subsamples were used as training data. All ten of the subsamples were utilized as testing data exactly once for each of the 10 iterations of the CV procedure. One estimate was created by averaging the 10 findings from the folds. We modified the four RF model hyperparameters based on Bayesian optimization to get the optimal model: the maximum number of features considered to split a node (Max features), the maximum number of trees to build (n estimators), the minimum sample number placed in a node prior to the node being split (min split), and the maximum number of levels for each decision tree (Max depth).

**Comment 4:** A description on the Bayesian optimization method in getting the optimal model is missing.

Response: Bayesian optimizer is used to tune parameters, and you can quickly find an acceptable hyperparameter value, compared with grid search, the advantage is that the number of iterations is less (time-saving), and the granularity can be very small. For example, if we want to adjust the regularized hyperparameters of linear regression, we set the black box function to linear regression, the independent variable is a hyperparameter, the dependent variable is linear regression in the training set accuracy, set an acceptable black box function dependent variable value, such as 0.95, the obtained hyperparameter result is a hyperparameter that can make the linear regression accuracy exceed 0.95.

**Comment 5:** Please clarify the reason of choosing the five elements (Rn, and WS, T, RH, and P) as the input variables.

Response: All of these predictors played a role in the H and LE fluxes to varying degrees. The most significant impact on H and LE changes came from net radiation, which was connected to the phenological characteristics of the plants. RH and T had a minor impact on the H and LE changes in terms of climatic parameters, which carried the information on the light-dependent reactions of H and LE fluxes. Particularly, WS and P had minimal impacts on the H and LE fluxes.

**Comment 6:** Please explain more about the total variable significance. What do they represent? Are these values normalized and comparable for different elements such as Rn, T, WS, RH?

Response: The possible drivers related to the H and LE are investigated to determine their respective contributions by the RF model. Rn, which accounted for 78% and 76% of the total variable significance, was the most crucial variable in regulating H and LE    With the highest r of 0.79 and 0.75, respectively, H and LE had the strongest positive association with Rn. RH, P, T, and T, WS, RH were three more prominent factors, with importance values of 7%, 5%, 4%, and 19%, 2.2%, 2%, respectively. The effects of WS and P were less noticeable and showed the lowest relative relevance, with values of 2% and 0.6%, respectively. All these elements such as Rn, T, WS, RH are normalized before the model starts training. When these elements are normalized, it ensures uniformity and comparability.

**Comment 7:** Figure 5 and Figure 6 : The performance of RF in the testing dataset is much better than that in the validation dataset, is there any possibility to improve the model's accuracy in the validation dataset? It seems that the fluxes are much larger in the training dataset

Response: Before model training, we eliminated data noise and other interference according to the standards of X(h) < (X 4) or X(h) > (X +4), where X(h) signifies the time series of the turbulence component, X is the mean across the averaging interval, and σ is the standard deviation. And in order to lessen the overfitting in this case, a 10-fold cross-validation (CV) procedure was used. All training data collected at the Shouxian location for the whole observation period was randomly divided into ten subsamples of equal size for the 10-fold CV tests. The remaining subsample was used as testing data, and nine of the ten subsamples were used as training data. All ten of the subsamples were utilized as testing data exactly once for

each of the 10 iterations of the CV procedure. One estimate was created by averaging the 10 findings from the folds. Through the above measures, the training,validation and testing results have been further improved.

[Figure]

**Figure 5. Scatter density plots of the observed and the RF-estimated H values, a) and d) for the training dataset, b) and e) for the validation dataset, and c) and f) for the testing dataset.   And a), b) and c) are in the period of rice, while d), e) and f) are in the period of wheat.**

[Figure]

**Figure 6. Same as Figure 5, but for LE.**

**Comment 8:** The model performance is largely improved when RH is excluded from the input, what is the reason behind it?

**Response:** This outcome is logical given that RH and H do not have a strong correlation, as a result, performance will be enhanced if RH is not included in the gap-filling processing pipeline. According to our findings, the RF model's performance may be greatly enhanced by excluding irrelevant meteorological elements from the study and choosing only those that have a significant impact on the variable. Our findings imply that in order to attain the best gap-filling accuracy, it is necessary to take into account both the advantages and disadvantages of ML-based models as well as the ideal input components.

**Comment 9:** The differences among the four ML methods are small. A discussion is needed to clarify more clearly their specialties and application situations.

Response:RF is a machine learning method that is quick, adaptable, and frequently used to analyze classification and regression jobs. This model can successfully evaluate highly dimensional and multicollinear data and is resistant to overfitting. The RF model provides a feature-selection tool to assist in determining the importance of the predictor. The contribution of each variable to the model, with important variables having a higher effect on the results of the model evaluation, is the definition of feature significance.
SVM is a data-oriented classification algorithm, and the basic model is to find the best separation hyperplane on the feature space so that the positive and negative sample intervals on the training set are maximum. Its advantages are that the kernel function can be used to map to a high-dimensional space; the use of the kernel function can solve the nonlinear classification; the classification idea is very simple, that is, to maximize the interval between the sample and the decision-making surface; the classification effect is better; and the nonlinear relationship between data and features is easy to obtain when the small and medium-sized sample size is large.
KNN is particularly suitable for multi-classification problems. Its advantage is that it is simple in thought, easy to understand, easy to implement; has no estimation parameters, no training;

High accuracy, insensitive to outliers.

GBDT can flexibly handle various types of data, including continuous and discrete values. With relatively few parameter adjustment times, the prediction preparation rate can also be relatively high. If the data dimension is high, the computational complexity of the algorithm will increase. Using some robust loss functions, the robustness to outliers is very strong.

LR is a statistical analysis method that uses regression analysis in mathematical statistics to determine the quantitative relationship between two or more variables that depend on each other.The results have good interpretability, can intuitively express the importance of each attribute in the prediction, and the calculation of entropy is not complicated.